# The Mitochondrial Dysfunction Hypothesis in Autism Spectrum Disorders: Current Status and Future Perspectives

**DOI:** 10.3390/ijms21165785

**Published:** 2020-08-12

**Authors:** Luigi Citrigno, Maria Muglia, Antonio Qualtieri, Patrizia Spadafora, Francesca Cavalcanti, Giovanni Pioggia, Antonio Cerasa

**Affiliations:** 1Institute for Biomedical Research and Innovation, National Research Council, IRIB-CNR, 87050 Mangone CS, Italy; luigi.citrigno@cnr.it (L.C.); maria.muglia@cnr.it (M.M.); antonio.qualtieri@cnr.it (A.Q.); patrizia.spadafora@cnr.it (P.S.); francesca.cavalcanti@cnr.it (F.C.); 2Institute for Biomedical Research and Innovation, National Research Council, IRIB-CNR, 98164 Messina, Italy; giovanni.pioggia@cnr.it; 3S’Anna Institute and Research in Advanced Neurorehabilitation (RAN), 88100 Crotone, Italy

**Keywords:** autism spectrum disorder (ASD), mitochondrial DNA (mtDNA), mitochondrial dysfunctions, next generation sequencing (NGS)

## Abstract

Autism spectrum disorders (ASDs) constitute a set of heterogeneous neurodevelopmental conditions, characterized by a wide genetic variability that has led to hypothesize a polygenic origin. The metabolic profiles of patients with ASD suggest a possible implication of mitochondrial pathways. Although different physiological and biochemical studies reported deficits in mitochondrial oxidative phosphorylation in subjects with ASD, the role of mitochondrial DNA variations has remained relatively unexplored. In this review, we report and discuss very recent evidence to demonstrate the key role of mitochondrial disorders in the development of ASD.

## 1. Introduction

Autism spectrum disorders (ASDs) are heterogeneous neurodevelopmental conditions, characterized by early-onset impairments in social communication and restricted, repetitive interests and behaviors [1]. The current estimated prevalence of ASD in the United States is one out of 54 children and worldwide ranges from 1 to 2% [2]. Individuals with ASD show marked heterogeneity at genetic, behavioral, aetiological, and pathophysiological levels. Several factors contribute to the heterogeneity in ASD, such as high rates of psychiatric disorders (such as attention deficit hyperactivity and anxiety), neurological conditions (such as seizures and sensory and motor system abnormalities), and comorbid medical conditions (e.g., sleep disorders, gastrointestinal dysfunction, immune system disorders) [1,3].

In the recent years, advanced genetic screening of nuclear DNA (nDNA), mainly focused on next-generation sequencing (NGS), has revealed an association between increased ASD risk with a significant number of gene mutations (>1000 genes), including Single Nucleotide Variants (SNV), Single Nucleotide Polymorphism (SNPs) and Copy Number Variations (CNV) [4,5,6]. These studies also show that the genetic vulnerability is not specific for ASD but it is shared with other neurodevelopmental conditions, further increasing the complexity [4,7] and suggesting the convergence on common dysfunctional pathways of the neuronal channel activity [8,9], neuronal communication [10,11], synaptic [12,13] and mitochondrial activities [14]. 

As concerns mitochondrial functioning, it has been suggested to play a putative role in ASD etiopathogenesis [14,15,16]. Indeed, mitochondrial dysfunction is not rare in children with ASD. László et al. [15] firstly reported, in a post-mortem study of an individual with ASD, alterations of bioenergetics metabolism with high levels of lactic acid, pyruvate, and serotonin. Subsequently, other post-mortem studies found high levels of the proteins implicated in the mitochondrial fission (FIS1 and DRP1) and a reduction of those involved in the mitochondrial fusion (MFN1, MFN2, OPA1). Alterations were identified in every I, II, III, IV, and V respiratory complex, as well as in the coenzyme-Q10, with decreased superoxide dismutase (SOD2) function and increased oxidative damage. Furthermore, these mitochondrial alterations have been found in brain regions mainly involved in the pathophysiological mechanisms of ASD, such as the cerebellum, prefrontal cortex, and temporal cortices [16,17]. 

However, the genetic basis of mitochondrial dysfunctions in ASD patients remains uncertain. Mitochondria have their genome (named mtDNA), which comprises a 16,569 bp circular molecule, including the genetic information necessary for the synthesis of 13 essential polypeptides in the mitochondrial respiratory chain [18]. It is worth noting that in each mitochondrion from 2 to 10 mtDNA molecules may be present, and each cell can contain several mitochondria [19]. Some evidence reported that children with ASD are more susceptible to mitochondrial dysfunctions with an increase in the replication and deletions of mtDNA, although other studies failed to replicate these findings [18,20]. Hence, to date, it is still unclear whether the mitochondrial dysfunction observed in ASD has a causal association with the condition or is instead a secondary event. Overall, identifying possible mitochondrial dysfunctions in children with ASD is very important since they may respond differently to specific treatments [21]. Indeed, Delhey et al. [22] have recently shown promising results by increasing the activity of complex I, complex IV, and citrate synthase, with the administration of specific mitochondrial supplements, such as fatty acid, folate, B12. 

For this reason, the present review is thought to provide recent advances in this challenging field of study for better understanding, as mtDNA alterations may be associated with the occurrence of ASD symptomatology. 

## 2. Results

As said in the Introduction, mitochondrial functional variation may play an important role in causing or increasing susceptibility to ASD. In this section, we report the main recent scientific papers supporting this hypothesis, and we summarize evidence provided by eight studies from 2016 to the present. In particular, we reported evidence of studies assessing the presence of mtDNA alterations in ASD patients to support (or less) the mitochondrial dysfunctions hypothesis of ASD. For this purpose and for facilitating comparisons, we summarized all relevant clinical and genetic findings in Table 1. 

### 2.1. Evaluating Pathogenic mtDNA Heteroplasmy Transmission in ASD 

By using exome-sequencing data sets, Wang and colleagues [23] investigated the connection between mtDNA variations (especially heteroplasmy) and ASD in simplex families. They analyzed data for 2709 individuals, including 903 trios (mother-proband-sibling) and compared mtDNA variation between autistic probands and non-autistic siblings. Among the 903 families, 191 were autistic probands (21.2%) and 182 non-autistic siblings (20.2%). On average, the overall mutation burden was not significantly different between autistic probands and their non-autistic siblings. The probands carried 0.25 mutations and non-autistic sibling 0.24 mutations. Moreover, in autistic probands, the heteroplasmic mutations were more located at non-polymorphic sites, contrasting to those in non-autistic siblings (chi-squared test, *p* = 0.0015). If we consider the mutations according to their functional annotations, autistic probands carried 52% more nonsynonymous mutations (0.084 per proband vs. 0.055 per sibling; *p* = 0.0028), and 118% more predicted pathogenic mutations compared to non-autistic siblings (0.041 per proband vs. 0.019 per sibling; *p* = 0.0016). In addition, autistic patients had about double mutations associated with the disease compared to non-autistic siblings.

The 265 mtDNA mutations were private, being detectable in only one child within a family. Carrying nonsynonymous private mutations was associated with an increased risk of ASD (Odds Ratio, OR: 1.87, *p* = 0.0055). Likewise, carrying mutations predicted to be pathogenic increases the risk of ASD by more than double (OR: 2.55, *p* = 0.0036). The results highlighted the role of mtDNA, suggesting that a possible mechanism underlying the metabolic pathophysiology of ASD could be the accumulation of mtDNA of high pathogenic potential during development.

### 2.2. NGS for Discovering mtDNA Variants

Patowary et al. [24] analyzed the mtDNA sequence derived from whole-exome sequencing in 10 ASD multiplex families. They sequenced four families with affected maternal cousins and six families with four or more affected siblings for a total of 35 individuals (25 male, 10 female). Out of 72 rare variants, five variants detected in four families met variants of interest (VOI) criteria. To assess interactions with nDNA) in these four families, 83 genes involved in respiratory chain complex (RCC) were analyzed by NGS. In a family comprising five affected siblings, two variants of interest, c.13528A>G (Tau398Asp) and c.13565C>T (*Ser410Phe*), in the *MT-ND5* gene were identified. The known variants were previously determined to impair mitochondrial function were. Moreover, in a cousin family, they identified two additional VOIs: a rare c.8896G>A (*Ala126Thr*) in *MT-ATP6* and a novel nDNA *Thr37Sser* NDUFS4 variant. Both VOIs are within mitochondrial RCC. No information about in vitro variant effects is known. However, the authors suggest that the combined effects of two VOI variants in the same family can result in mitochondrial dysfunction, as already reported by Anderson et al. [31], who demonstrated that common mitochondrial genetic variants may influence the risk of sporadic ischemic stroke. These findings provide further support for the role of mitochondria in ASD.

### 2.3. The Occurring of mtDNA Deletions in ASD

Varga and colleagues [25] investigated correlations between mtDNA changes and alterations of genes associated with mtDNA maintenance in ASD patients. Sixty patients with ASD and sixty healthy individuals were screened for common mtDNA variations. To identify single and multiple mtDNA deletions, long-range PCR was performed. Three common variations, *m.3243* A>G, *m.8993* T>C/G, and *m.8344* A>G, were investigated by Restriction Fragment Length Polymorphism (RFLP). NGS was used to investigate the most well-known ASD-associated genes (101 genes) and 51 genes responsible for intergenomic communication disturbances. NGS analysis was performed in small subgroups of ten cases with mtDNA deletion, seven ASD non-mtDNA deletion cases and six healthy controls. Pathogenic and likely pathogenic mutations from NGS data were validated by Sanger sequencing, and segregation analysis was performed within individual families. Mitochondrial deletions were identified in 16.6% of ASD patients. Nine of ten cases with mtDNA deletions carried rare SNVs in ASD-associated genes (one of those was pathogenic). In the intergenomic panel of this cohort, only one likely pathogenic variant was present. In patients with mitochondrial disease in genes responsible for mtDNA maintenance, pathogenic mutations were detected more frequently than those found in patients from the mtDNA deletions. These authors claimed that the mtDNA deletions are more frequent in ASD, but they coexist either with other ASD-associated genetic risk factors or with alterations in genes responsible for intergenomic communication.

### 2.4. Overexpression of Mitochondrial Gene in ASD

Park and colleagues [26] carried out a study on two Korean siblings with ASD [26]. They collected peripheral blood samples from the five family members and examined genomic DNA using whole-exome sequencing (WES). A rare homozygous c.790C>T (*His264Tyr*) variation in the mitochondrial transcription factor B2 (*TFB2M*) gene was identified. *TFB2M* is an essential protein in mitochondrial gene expression. To better clarify the possible role of the *TFB2M* variation in the pathogenesis of ASD in this family, the authors investigated the characteristics of primary cultured fibroblasts from the proband and his father by skin biopsy. No significant differences in mRNA and protein levels were observed between the patient and his father. However, mRNA levels of the mitochondrial genes encoding components of the mitochondrial oxidative phosphorylation system complexes I, III, IV, and V were significantly higher in the patient than in the father. These results suggested that the variation in *TFB2M* c.790C>T does not influence the expression of the *TFB2M* gene but causes a marked increase in the transcription of mitochondrial genes involved in the mitochondrial oxidative phosphorylation system. Patient fibroblasts revealed increased transcription of mitochondrial genes and mitochondrial function in terms of ATP, membrane potential, oxygen consumption, and reactive oxygen species. In addition, a molecular dynamics simulation of the *TFB2M* variant protein suggested an increase in the rigidity of the hinge region, which may perturb the loading and/or unloading of *TFB2M* on target DNA. These findings suggest that an increase of mitochondrial gene expression and subsequent mitochondrial dysfunction may be associated with the pathogenesis of ASD in Korean patients.

### 2.5. Mitochondrial Dysfunction in ASD with Respect to Intellectual Disability

It has been demonstrated that frequent medical comorbidities can occur in ASD patients, such as epilepsy, sensory abnormalities, sensory/motor deficits, sleep abnormalities. These clinical conditions may be grouped under the umbrella term commonly associated with mitochondrial disorders (CAMDs), since impaired mitochondria affect most organ systems, producing a wide spectrum of phenotypes. The purpose of the Valiente-Palleja et al. study [27] was to evaluate the presence of CAMDs and mitochondrial DNA (mtDNA) alterations in ASD and patients with an intellectual disability (ID). The authors investigated 122 subjects who presented ID and ASD (ID-ASD group); 115 subjects who presented ID but not ASD (ID group) and 112 healthy controls (HC). Mitochondrial genome sequences from patients have been generated by NGS mtDNA-targeted approach using the Ion Torrent Personal Genome Machine (PGM) from ThermoFisher Scientific, whereas the presence of mtDNA mutations was evaluated using the MToolBox pipeline. By using two mtDNA genes (*MT-ND1* and *MT-ND4*) and Nuclear RPPH1 gene (corresponding to the RNase P enzyme) as a single-copy nuclear gene reference, the authors were able to calculate the mtDNA copy number by quantitative real-time PCR. The mtDNA in ASD and ID subjects was significantly lower than HCs in the gene *MT-ND1* and *MT-ND4*. The authors were also able to identify 49 putative pathogenic variants with a level of heteroplasmy in more than 60% of the enrolled population: 8 missenses, 29 in the rRNA and 12 tRNA variants. Moreover, it was found that a total of 28.6% of ASD and 30.5% of ID subjects present at least one possible pathogenic mtDNA mutation. In this work, the authors demonstrated that the conditions commonly associated with mitochondrial disorders are commonly present in ASD and ID patients, also correlating with the presence of low mtDNA content and putative pathogenic mtDNA mutations.

### 2.6. Extracellular Vesicles in ASD Children Contain mtDNA

Tsilioni et al. [28] sought to evaluate the possible role of the nucleic acid and protein composition of the Extracellular Vesicles (EVs) that could be the trigger of the brain inflammation in the ASD disorder. The generation of the EVs begins from the cell when multivesicular bodies) fuse with or can be released from the plasma membrane. To evaluate the EVs function, the authors analyzed the serum of ASD children (*n* = 20, 16 males and four females, 4–12 years old) and unrelated age- and sex-matched healthy controls (*n* = 8, six males and two females, 4–12 years old). The EVs were characterized by determining the CD9 and CD81 as membrane-associated markers by Western blot analysis, whereas the size and morphology were analyzed using transmission electron microscopy (TEM). The total DNA coming from the EVs isolated from the serum was extracted using Qiagen DNA Microextraction kit. Mitochondrial-specific DNA for 7S (mtDNA7S) was isolated and quantified by real-time PCR (RT-PCR) using TaqMan gene expression assays. Tsilioni and colleagues described both the presence of EVs in the serum of the ASD patients and that these EVs contain a significant amount of mtDNA compared to the normal controls. The authors also suggested that the mtDNA coming from the EVs can be used as an alarmin and brought to pro-inflammatory mediators from immune cells. In ASD, mtDNA may represent an “innate” pathogen that can be protected from the degradation inside EVs and can arrive in the microglia through the blood–brain barrier and brain lymphatics system.

### 2.7. Association between mtDNA Haplogroup-Linked Functional Variants and Risk of ASD 

Usually, the types of clinically relevant mtDNA variations can be divided into three categories: (1) single pure mtDNA allele: polymorphisms associated with ancient adaptive haplogroup considered as homoplasmic; (2) a mixture of two mtDNA alleles: recent deleterious polymorphisms in the female germline within the past 10 generations (both homoplasmic or heteroplasmic); and (3) somatic mutations of the developmental that are invariably heteroplasmic. Since the genetic complexity of ASD and the association with mitochondrial bioenergetic defects, mtDNA haplogroup variations can contribute to the modulation of the ASD risk. Chalkia D. et al. [29] investigated 1624 patients with ASD (1299 male and 325 female) and 2417 healthy parents and siblings [29]. The authors analyzed and used the data about the mtDNA single nucleotide polymorphisms to determine the mtDNA haplogroups of the individuals with ASD. The mtDNA SNVs were determined using the Illumina HumanHap 550 array with a genome wide association study analysis approach in the patients and their families. Data analysis was performed using PLINK software. They found that numerous mtDNA haplogroups were distributed across different clusters in ASD patients. The authors suggest that mitochondrial haplogroups and their associated functional variants can contribute to increasing the ASD risk. Indeed, the linkage analysis of mtDNA haplogroups in ASD supports a mitochondrial component to the causes of ASD.

### 2.8. Alterations of Mitochondrial Biology in the Oral Mucosa of Individuals with ASD

In this paper, the authors proposed a non-invasive procedure to assess mitochondrial DNA function in children with ASD, through the use of oral mucosa [30]. Generally, the evaluation of mitochondrial function requires an invasive biopsy, whereas the employment of this non-invasive procedure is a valid alternative, since the oral mucosa derives from the embryonic ectoderm, making them a relevant tissue for ASD studies. The muscular biopsy was widely considered the gold-standard for evaluating mitochondrial dysfunction. However, the use of this relative invasive technique can be complicated in children with ASD. The use of oral mucosa to detect mitochondrial dysfunction has been already validated by Goldenthal et al. [32]. They evaluated in buccal swap specific mitochondrial dysfunction in patients with defects in biopsied skeletal muscle, giving evidence that the oral mucosa can be used to detect pathogenic mtDNA mutations. In an additional paper, Goldenthal at al., [33] extended their findings, identifying extensive abnormalities in the activity of mitochondrial respiratory complex I and IV in a buccal swab analysis of ASD children. They proposed, for the first time, that the respiratory chain activities can be considered as a noninvasive biomarker of mitochondrial enzyme dysfunction in ASD. The authors investigated 24 children with and without ASD. The sample collection process was carried out by a non-invasive procedure to obtain the nucleid acid coming from the oral mucosa cells of the children. After the extraction and quantitation of the nucleic acids, the authors carried out a series of different analyses: the gene expression analysis of seven genes (*HIGD2A*, *SOD2*, *DRP1*, *FIS1*, *MFN1*, *MFN2*, and *OPA1*) and the analysis of the whole mitochondrial DNA and proteins. The results showed that there is a significant increase in the mtDNA levels in the oral mucosa of ASD children, which correlated with mitochondrial dysfunctions. Again, the authors demonstrated that 75% of children with ASD and 92% of the population of children with HC could be identified as being heterozygous for the *Ala16Val-SOD2* (A/V) polymorphism. This polymorphism can reduce enzymatic activities and increase oxidative stress. Finally, they demonstrated an increased level of genes involved in the mitochondrial fission/fusion mechanisms, such as *MFN1*, *MFN2* and *OPA1*. The authors concluded highlighting the role of this kind of analysis as an alternative for studying gene expression associated with the development of ASD in humans since the oral mucosa has an embryonic origin.

## 3. Discussion

The human brain constantly consumes energy to maintain its function and demands exceptional production in the mitochondria [34]. The mitochondrial dysfunction hypothesis has been proposed [14] to explain the transversal mechanisms underlying biological processes of neurodevelopmental deficits in ASD. The main findings reported in the selected articles suggest the involvement of mtDNA in a subgroup of individuals with ASD. Indeed, an accumulation of mtDNA mutations of high pathogenic potential during development may be a key mechanism underlying the pathophysiology of ASD. 

The evidence provided so far describes either variations directly involving the nucleotide sequence of mtDNA [24,25,27] or variations concerning the quantity of the same mutated mitochondria in the cell [23] (for a complete review of the main findings, see Table 2). The most important variations in the nucleotide sequence fall mainly in mitochondrial genes that encode proteins involved in cellular respiration [24,27], a set of metabolic processes occurring in the cells of organisms useful to convert chemical energy from oxygen molecules into adenosine triphosphate. As stated by Park et al. [26], because mtDNA encodes 13 essential subunits of the respiratory chain, the integrity of mtDNA and the expression of genes within mtDNA are crucial for maintaining the oxidative phosphorylation system. Any alterations in these processes can cause perturbation in cellular energy production, which may potentially lead to a disease state [35]. It has been proposed that disorders of the mitochondrial respiratory chain may be associated with a high frequency of metabolic abnormalities in pediatric neurology [36]. As concerns ASD children, it has been suggested as a key mechanism involved in neuronal cell damage. Mitochondrial respiratory electron transport chain is formed by five multi-subunit enzymes that is different both in structure and in function, playing a crucial role in the production of adenosine triphosphate (ATP). These multi-subunit enzymes participate in the generation of the proton gradient in the mitochondrial intermembrane space, necessary to transform adenosine diphosphate (ADP) into ATP using the ATP synthase. In the brain, almost 90% of this energy is made by the respiratory electron transport chain [35]. The basal functions of the respiratory electron transport chain are essential to maintain the structure and function of neurons. Therefore, abnormalities in the respiratory electron transport chain during the first stage of development may be involved in the etiology of neurodevelopmental disorders. Interestingly, dysfunctions have been found in nuclear genes that encode for proteins specifically involved in mitochondrial fission and fusion, such as *MFN1*, *MFN2*, *OPA1*, and that are involved in mitochondrial dynamics [30]. Mitochondria are dynamic organelles, constantly changing their morphology or shape by undergoing fission (fragmentation) and fusion (elongation). Mitochondrial fission involves the division of a single mitochondrion into two fragmented mitochondria by the fission proteins (DRP1) [37] human fission protein 1 (hFis1) [38], mitochondrial fission factor (Mff) [39] and the mitochondrial dynamics proteins of 49 kD (MiD49) and 51 kD (MiD51) [40]. Mitochondrial fission is essential for cell division, for facilitating intracellular energy distribution and is required to remove damaged mitochondria by mitophagy. This mechanism is characterized by the tethering of two adjacent mitochondria, mediated by two GTPase proteins, mitofusin 1 (Mfn1) [41] and mitofusin 2 (Mfn2) [42], which determine the fusion of the outer membrane mitochondrial (OMMs). Subsequently, the optic atrophy 1 (*OPA1*) protein mediates the fusion of the inner mitochondrial membrane (*IMMs*), resulting in the sharing of mitochondrial matrix material and the formation of a single elongated mitochondrion. Recently, it has been reported that this mitochondrial dynamism can be considered an adaptive mechanism characterized by greater energy demand and a persistent inflammatory oxidative state following a compromised detoxifying pathway. In fact, oxidative damage is present both in the fibroblasts of ASD patients and at the peripheral and central nervous system levels. In this perspective, Pecorelli et al. [43] have recently suggested the use of antioxidants and an appropriate diet as a more effective therapy for reducing the symptoms of the disease.

Other authors [29] supporting the mitochondrial dysfunction hypothesis evaluated the differences in the different mitochondrial haplogroups between children with ASD and control subjects. These authors suggested a mitochondrial-mediated mechanism to explain how the mtDNA haplogroups are likely to play an interesting role in the genetic etiology of autism, through the interaction with additional genetic variants in the nuclear genome.

ASD can often occur with other medical comorbidities (i.e., sleep disorders, sensory abnormalities, epilepsy, gastrointestinal disturbances), which are commonly referred to as CAMDs. This relationship is generally correlated with low mtDNA content and/or putative pathogenic mtDNA variants, demonstrating the strong involvement of the mtDNA in the complex phenotypes that coexists in ASD [27]. Indeed, comparing individuals with ASD with respect to those with intellectual disabilities, Valiente-Pallejà et al. [27] found several common variants confirming that mitochondrial genetic variations cannot distinguish between the two clinical phenotypes. 

Overall, what emerges from this review is that the exact genotype–phenotype relationship for all the reported genetic variants remains to be established. Today, it is still difficult to determine if mitochondrial dysfunctions are associated as moderators or mediators to the etiology of ASD and it is very likely that mitochondrial alterations are common in some neurodevelopmental conditions. Indeed, one should bear in mind that many highly penetrant genetic causes of ASD have been identified, but, as suggested by Schaaf et al. [44], “the underlying genetic background of 70% of cases remains still unexplained”. This concept has also been developed by Mayers et al. [45] who recently provided a divergent interpretation, because the pathogenic variants found in ASD patients are also present in different clinical disorders with overlapping symptoms such as ID, epilepsy, schizophrenia, and other neuropsychiatric conditions.

Regardless of the mechanism that ultimately determines mitochondrial dysfunction in ASD, it appears reasonable being able to intervene therapeutically as early as possible, already during the neonatal period, by administering mitochondrial supplements and vitamins, as recently indicated by Lehmann and McFarland [46]. This intervention could be addressed to infants with mitochondrial dysfunctions detectable by analyzing cellular samples obtained by non-invasive techniques such as the buccal swap and urine. 

## 4. Conclusions

The purpose of this review is to summarize and compare recent studies evaluating the impact of mtDNA variations in the pathophysiological mechanisms of ASD patients. The higher prevalence of mitochondrial dysfunctional phenotypic signature in ASD subjects would further strengthen its involvement in pathophysiological mechanisms. In addition, understanding the mitochondrial contributions to autism could be rapidly translated into new diagnostic tools or tailored therapeutic interventions for specific dysfunctions (protein, fission and fusion and cellular respiration). 

The use of appropriate genetic tests as the whole genome sequence is important for a definitive clinical diagnosis to define the risks and progression of the disease and the other family members [47]. ASD is probably the result of a genetic predisposition and epigenetic mechanisms. The work of Stathopoulos et al. [48] showed how the methylation of mitochondrial DNA plays an important role in the etiology of ASD. Therefore, understanding the mitochondrial mechanisms involved in ASD could be important for the identification of new targets in the treatment of pathology. Regarding this topic, a very recent study evaluated the mitochondrial bioenergetics in the BTBR mouse model of ASD [49]. The authors investigated changes in mitochondrial morphology, which can directly influence mitochondrial function. They found that BTBR mice had altered mitochondrial function and exhibited smaller more fragmented mitochondria compared to C57BL/6J controls. In addition, they also identified activating the phosphorylation of two fission proteins, pDRP1S616 and pMFFS146, in BTBR mice, consistent with the increased mitochondrial fragmentation that they observed. It is already known that the ketogenic diet (KD), a high-fat, low-carbohydrate and low-protein diet, reduces autistic behaviors in both humans and rodent models of ASD. Intriguingly, it has been found that the supplementation of KD to the BTBR mice improved both mitochondrial function and morphology. Furthermore, the KD decreased pDRP1S616 levels in BTBR mice, likely contributing to the restoration of mitochondrial morphology. These data give further evidence that impaired mitochondrial bioenergetics and mitochondrial fragmentation may contribute to the etiology of ASD and that these alterations can be reversed with KD treatment.

All evidence provided so far highlights the importance of exploring and providing firm conclusions on the association of these genes with ASD and their suitability as ASD biomarkers.

## Figures and Tables

**Table 1 ijms-21-05785-t001:** Main findings of studies assessing the impact of genetic variations in mtDNA in patients with Autism spectrum disorder (ASD). For a list of abbreviation see Abbreviations section.

References	Samples	Aim	Methods	Main Findings	Conclusions
Wang et al., 2016 [23]	903 ASD families & 2709 healthy individuals	Cross-sectional study to compare mtDNA variations between autistic proband and non-autistic siblings	Whole-exome sequencing data were obtained from the National Database for Autism Research. The sequencing data were generated by three genome centers.	The general mutation burden was not significantly different between groups. Heteroplasmic mutations were more located at non-polymorphic sites in autistic probands with respect to autistic probands carried.	Autistic probands vs. no autistic siblings: no differences in overall mutation burden. Heteroplasmic mutations were more located at non-polymorphic sites.
Patowary et al., 2017 [24]	10 ASD multiplex families for a total of 35 individuals	Cross-sectional study to investigate mtDNA changes in subjects with familial autism and interactions; between mtDNA and nDNA	Deep DNA sequencing with next-generation sequencing (NGS) to study mtDNA and nDNA	In one of the families two variants in *MT-ND5* gene have been identified. In another family two VOIs; mtDNA variant in *MT-ATP6* and nDNA variant in *NDUFS4*	The results provide further evidence of the role of mitochondria in ASD and confirm that whole-exome sequencing is a rapid tool to analyze mtDNA, an important step to better understand the role of mitochondria in autism
Varga et al., 2018 [25]	60 ASD 7 patients with Mitochondrial Disease (MD) 60 healthy control	Cases-controls study to investigate mtDNA changes and alterations of genes associated with mtDNA maintenance in ASD	Longe range Polymerase chain reaction(PCR) for mtDNA deletions; RFLP for three common variations in mtDNA; NGS using two panels containing 101 ASD-associated genes and 51 genes responsible for intergenomic communication disturbances, respectively	Mitochondrial deletions were identified in 16.6% of patients with ASD. Out of 10 cases with mtDNA-del analyzed by NGS panel with 101 genes, 9 (90%) carried rare Single Nucleotide Variants (SNVs), but only one was pathogenic. In the panel one likely pathogenic variant MD patients > Variant of Uncertain Significance (VUS) than in patients from the model-ASD or other comparison groups	mtDNA deletions are more frequent in ASD than in healthy individuals, but they are not isolated, but often in association with other ASD-associated genetic risk factors or with variations in genes responsible for intergenomic communication
Park et al., 2018 [26]	5 ASD family members	Cross-sectional study to investigate the TFB2M variant in the pathogenesis of ASD.	Whole-Exome Sequencing (WES) with NGS; primary cultured fibroblasts; Western blotting; *TFB2M* expression constructs	A variation in homozygous state was identified in *TFB2M* gene in two Korean siblings with ASD. This variation caused increased transcription of mitochondrial genes and the mitochondrial function. Structural changes of variant *TFB2M* could alter the unloading/loading of *TFB2M* to DNA target.	The increasing of mitochondrial gene expression and subsequent enhancement of mitochondrial function (increased oxidative stress during brain development) may be responsible for the pathogenesis of ASD.
Valiente-Pallejà A. et al., 2018 [27]	122 with ID and ASD; 115 with ID but not ASD; 112 healthy controls	Cross-sectional study to test the presence of Conditions associate with Mitochondrial disorders (CAMDs) and mitochondrial DNA (mtDNA) alterations in ASD and ID	Next generation sequencing mtDNA-targeted approach using the Ion Torrent Personal Genome Machine (PGM)	The mtDNA was significantly lower in ASD than healthy controls (HCs), considering the genes *MT-ND1* and *MT-ND4* genes. 49 putative pathogenic variants were identified with a heteroplasmy level higher than 60%: 8 missense, 29 rRNA and 12 tRNA variants	Low mtDNA content and putative pathogenic mtDNA mutations in subjects with ASD and ID correlate with a high frequency of conditions commonly associated with mitochondrial disorders
Tsilioni I. et al., 2018 [28]	20 Caucasian children (16 males and 4 females, 4-12 years old) with ASD 8 normotypic controls (6 males and 2 females, 4–12 years old)	Cases-controls study to identify the presence of the mtDNA in the Extracellular Vesicles (EVs) in patients with ASD	The EVs were isolated from blood using ExoEasy Qiagen, characterized by CD9 and CD81 markers, while their morphology and size were analyzed by TEM. The total DNA coming from the EVs was extracted from EVs using Qiagen DNA Microextraction kit. mtDNA 7S was detected and quantified by real-time PCR using TaqMan gene expression assays	The serum of the patients affected by ASD contains a significant amount of mtDNA compared to the normal controls.	The mtDNA coming from the EVs can represent an alarmin inducing pro-inflammatory mediator secretion from immune cells
Chalkia D. et al., 2017 [29]	1624 patients with ASD (1299 boys and 325 girls); 2417 healthy parents and siblings.	Cases-controls study to show that mtDNA haplogroups can contribute to risk for ASD	The mtDNA SNVs were determined using the Illumina HumanHap 550 array and statistical analysis was performer using PLINK software	mtDNA haplogroup variation significantly correlates with risk for ASD.	The linkage analysis of mtDNA haplogroups with ASD supports the hypothesis that mitochondrial functional variation is an important risk factor for ASD.
Carrasco M. et al., 2019 [30]	12 children with ASD (11 male, 1 female) 12 children without ASD	Descriptive transactional and non-probabilistic design to study mtDNA in the oral mucosa of children with ASD	mtDNA analysis performed with quantitative PCR (qPCR). Genotyping of the Ala16Val-SOD2 Single Nucleotide Polymorphism (SNP) polymorphism was performed by PCR using the DNA from the oral mucosa sample and a “Tetra-Primer ARMS-PCR” assay. For the immunodetection of respiratory complexes, 20 ug of total protein samples from oral mucosa were studied. Total protein oxidation was measured with the Oxyblot Detection Kit (Millipore)	The authors found a significant increase in the mtDNA levels in the oral mucosa of Chilean, which is correlated with mitochondrial dysfunction. The authors found an increased level of genes involved in the mitochondrial fusion/fusion mechanisms, such as *MFN1, MFN2* and *OPA1*	The oral mucosa sample may be considered a reliable tissue to study mtDNA levels in children with ASD.

**Table 2 ijms-21-05785-t002:** Summary of the main findings of the Mitochondrial Dysfunction Hypothesis in ASD.

Authors	Main Findings
Wang et al., 2016 [23]	No differences in overall mutation burden between ASD and normal controls. The heteroplasmic mutations were more located at non-polymorphic sites.
Patowary et al., 2017 [24]	Two mtDNA variants in *MT-ND5* and *MT-APT6* genes of ASD patients.
Varga et al., 2018 [25]	Deletions in mtDNA are frequent, but they can exist together with another ASD-associated risk factor.
Park et al., 2018 [26]	ASD patients are characterized by an overexpression of mitochondrial *TFB2M* gene, with a consequence in increased transcription of mitochondrial function in terms of ATP, membrane potential and oxygen consumption.
Valiente-Pallejà A. et al., 2018 [27]	The lower presence of mtDNA in ASD with respect to HCs correlates with a high frequency of CAMDs.
Tsilioni I. et al., 2018 [28]	The mtDNA can be found in the EVs coming from the serum of the patients with ASD.
Chalkia D. et al., 2017 [29]	The mtDNA haplogroups as a risk factor for ASD.
Carrasco M. et al., 2019 [30]	The oral mucosa of patients with ASD can contain mtDNA with a significant increase for the genes: *MFN1*, *MFN2* and *OPA1*.

ASD: autism spectrum disorders; mtDNA: mitochondrial DNA; CAMDs: clinical conditions commonly associated with mitochondrial; EVs: extracellular vesicles.

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
