# Peer review of "The Mitochondrial Dysfunction Hypothesis in Autism Spectrum Disorders: Current Status and Future Perspectives"

_ijms, 2020, doi:10.3390/ijms21165785_

Round 1
Reviewer 1 Report
This is a well-organized paper on the current status and future prospects on ASD. The authors suggest as the role of mitochondrial DNA variation could be offered a new point of view to the characterization of ASD patients.
In its entirety, the paper is well organized and the display of the chosen works is well performed.
However, it would be appropriate to include more recent work in the discussion to underline the future prospective (for example Pecorelli et al., 2020 The FASEB Journal; Kreiman & Boles Semin Pediatr Neurol. 2020; Stathopoulos et al., 2020 Autism Res).
Author Response
As suggested by this reviewer these additional references have been included in the main text. See pag 9 and 10
Reviewer 2 Report
The review entitled " The Mitochondrial Dysfunction Hypothesis in Autism Spectrum Disorders: Current Status and Future Perspectives" by Citrigno et al. is an excellent piece of work that summarizes the state of the field in one of the most exciting areas of autism research, mitochondrial activity. The review is well written and covers the most relevant works in this regard. I would like to see it published. However, this reviewer feels that the discussion misses some speculation about the possibility of targeting mitochondrial function as a potential treatment for ASDs. This work will also benefit from the inclusion of an overall schematic to describe a model/interpretation of the exposed concepts in the text. Minor comment: Glessner et al. (PMID: 19404257) should be mention in the text.
Author Response
- Following reviewer’s suggestion, we now provide new speculations on the mitochondrial mechanisms as important new targets for treatment of ASD. See pag. 10
- We now include a new table 2 to summarize the main findings of this mini-review. The multicenter study made by Glessner and colleagues has been included.